# Barriers to Formal Help-Seeking Behavior by Battered Turkish Women According to Sociodemographic Factors

Abdulkadir Keskin [1,*] and Filiz Karaman [2]

1   Dumlupınar Mahallesi D-100 Karayolu, İstanbul Medeniyet University Faculty of Political Science Güney Yerleşkesi A Blok 2. Kat, 34734 İstanbul, Turkey
2   Department of Statistics, Davutpaşa Campus, Yıldız Technical University, 34220 İstanbul, Turkey; fkaraman@yildiz.edu.tr
*   Correspondence: abdulkadir.keskin@medeniyet.edu.tr

**Abstract:** Violence against women is a significant sociological problem that negatively affects society. Although violence against women is widespread worldwide, the help-seeking behavior of women exposed to violence remains underdeveloped. In this study conducted in Turkey, the formal help-seeking behavior of women exposed to violence was studied according to sociodemographic factors. Data were obtained from surveys on domestic violence against women from the Turkish Statistical Institute (TURKSTAT), which was held in 2008 and 2014. Chi-square and binary logistic regression analyses are used in this study. The dependent variable is determined as whether women who had experienced violence sought help from formal institutions. In order to explain the dependent variable, education, age group, region, and sociodemographic variables are used. According to the results, as the education level of women exposed to violence increases, help-seeking behavior through official means also increases. In addition, women with a personal income are more likely to seek formal help than those without, and the development of the sociocultural region inhabited affects the formal help-seeking behavior of women exposed to violence. Between 2008 and 2014, legal regulations on women's rights in Turkey were seen to positively affect formal help-seeking behavior. Although the formal help-seeking behavior of women subjected to violence in 2014 increased significantly compared to 2008, this improvement is not sufficient.

**Keywords:** intimate partner violence; formal help-seeking behavior: Turkey; barriers to help-seeking behavior; Istanbul Convention

## 1. Introduction

Studies show that intimate partner violence (IPV) against women is widespread in developed and developing countries [1–3]. Violence is one of the leading violations of rights against women and is prevalent worldwide, regardless of region. Campell et al. define violence against women as "all verbal, physical and sexual assaults that violate a woman's physical body, sense of self and confidence, regardless of age, race, ethnic origin or country" [4]. Different forms of violence exist in the relationship between partners, such as physical, economic, verbal, emotional, and sexual, which are all common in bilateral relationships [5].

On the other hand, Joachim [6] states that violence against women is also a significant public health and human rights problem. Accordingly, domestic violence can lead not only to mental health problems such as trauma, depression, and anxiety disorders in victims but also to intergenerational violence and children's mental health disorders [7–9]. In this context, it can be stated that violence against women, as a public health problem, can cause mental disorders not only in women subjected to violence but also in all family members.

In a report on violence against women published by the World Health Organization in 2013, it was revealed that women were subjected to violence in every period of their lives. According to the published report, as a result of the research conducted on women between

the ages of 15 and 49, 13% to 61% of the women had been subjected to physical violence by their spouses at least once in their lives. Between 6% and 59% of the women were exposed to forced or attempted sexual intercourse by their spouses at least once in their lifetime, and 1% to 28% were exposed to physical violence by their spouses during pregnancy [1]. According to Güvenç et al. [10], in a study conducted in Turkey, the rates of violence against women in Turkey were revealed between 13% and 78%. In another study conducted in general (research on IPV, which was held in 2014 in Turkey), in any period, the proportion of women who stated they suffered physical violence in the previous year of their life was between 8.2% and 35.5%. In the same study, 12% of women in Turkey reported sexual violence, and 37.5% reported physical or sexual violence. Within the scope of the study, it was determined that 43.9% of women were subjected to emotional violence/abuse in any period of their lives and 30% to at least one form of economic violence/abuse (Directorate General of Women's Status) [11].

However, although IPV is common, most women subjected to violence do not seek help due to specific reasons [12]. Victims of IPV internalize the violence and conflict they are exposed to within their family and transform their suffering into the perception of a normal situation [13].

It was found that most women previously subjected to IPV did not seek help in formal or informal ways [12,14]. In general, help-seeking behavior can be defined as the process of applying to official institutions (police, gendarmerie family court, lawyer, bar, etc.) [15]. However, women subjected to violence may also seek help in informal ways (family members, relatives, friends, etc.) to protect themselves from violence [16,17].

It has been determined that women tend not to seek help for a number of sociocultural reasons despite being subjected to violence from their partners and maintain relationships where they are exposed to violence. Some researchers have tried to explain this situation with psychological theories [18–21].

Studies conducted in this context reveal the reasons women subjected to IPV continue their relationships: frequent exposure to violence, not believing that they will receive support, not wanting to disband their family, fear of social stigma, not knowing what to do, perceiving IPV as normal, emotional attachment to their partner, being economically incapable, not receiving support from their family and environment, fear of separation from their children, hope that their partner will change, lack of shelter, and society's attitude toward divorced/lonely women [11,22,23].

In addition, the sociocultural environment affects women's attitudes toward IPV. Accordingly, women subjected to IPV believe that their experiences with violence cannot be shared with strangers and such incidents can only be shared with reliable people such as family and relatives able to hide information. For example, studies on the help-seeking behavior of Asian women subjected to IPV have shown that some cultural values, along with IPV, present an obstacle, preventing women from talking with others [24,25]. Niaz found that most women in Asian countries believe that family matters, including IPV, are considered private and should be kept as such [26]. Lettiere and Nakano, in their study in Brazil, found that women exposed to violence initially tried to hide the violence they experienced from the environment in which they lived by first confiding in family members and later to religious organizations [27].

Many scientific studies and reports prepared in Turkey reveal a serious problem with IPV [28–30]. Almost all of these studies focus on violence against women and its causes, but a limited number of them mention women's help-seeking behavior. In addition, these studies were not conducted directly on women's help-seeking behavior. For example, in the study conducted by Altınay and Arat, women were asked how they react when subjected to violence by their husbands [31]. According to the results of the study, the women responded to the violence they were subjected to by being quiet, reacting physically or verbally, asking for a divorce, leaving home, and crying; however, just 5% of women said they decided to seek help through the police. In another study conducted on a provincial basis, it was determined that the prevalence of physical violence was 34%, and less than

half of these women went to the police [32]. On the other hand, according to the results of Turkey's official and most comprehensive study examining women's help-seeking behavior by Ergöçmen et al. [33], women subjected to violence sought help informally rather than formally. They found that as the dose of violence increased, formal help-seeking behavior also increased.

Studies on the help-seeking behavior of women subjected to violence have generally been conducted in developed countries, and these studies are limited in developing countries [34]. The fundamental aim of this study is to try to explain the help-seeking behavior of women subjected to violence according to socioeconomic and demographic factors. This study will fill in the missing gaps in the area of the help-seeking behavior of women who experience violence in Turkey. Turkish Statistical Institute (TURKSTAT) data structures were used in this study. This study was conducted with the most comprehensive and reliable data representing Turkish society.

The main objectives of this study are to try to explain the factors that affect the help-seeking behavior of women subjected to violence, such as education level, residence region, and employment status, and initiate change in the legal framework, Istanbul Convention, which may impact the help-seeking behavior of women in Turkey.

## 2. Materials and Methods

### 2.1. Data and Statistical Analysis

The sample survey of women exposed to domestic violence in Turkey was a weighted, stratified, and multistage cluster sample. Turkey is composed of 12 regions and 81 provinces. Sampling selections were made using the probability proportional to size (PPS) method within the strata [35]. It aimed to meet with a total of 24,048 households in 2008 and 15,072 households in 2014. In 2008, 17,168 household interviews among eligible households were conducted face-to-face with 12,795 women, with a response rate of 86%. In 2014, 11,247 household interviews were conducted face-to-face with 7462 women, with a response rate of 83.3%. In total, 18,709 successful interviews were conducted between 2008 and 2014. A suitable woman from each household was selected according to the method of Kish [36]. Two forms were used as household questionnaires and women's questionnaires to obtain the data. The questionnaire of the study was used by updating the questionnaire used by the World Health Organization's study, "Multi-Country Study on Women's Health and Domestic Violence against Women" [1].

The analysis used in this article was made using quantitative data from the National Survey of Domestic Violence against Women in Turkey. This study provides the most current and comprehensive data on IPV within families in Turkey. It is also a cross-sectional study conducted at the national level. The data were collected using a questionnaire through interviews with women. Survey datasets belonging to the Turkey Statistical Institute (TSI) were taken with special permission.

Firstly, Pearson's chi-square significance test was used to examine the relationship between women subjected to violence applying to official institutions and the sociodemographic factors affecting these applications. Subsequently, the tendency of women subjected to violence to seek help from any official institution or organization was analyzed by creating a logistic regression model with variables such as age, education, personal income, and region of residence. In several studies, although many women were subjected to violence, it was observed that they did not share their situation with their immediate surroundings, especially family members and close friends [37,38]. This variable was also included in the model of the study to determine whether women who share the violence they experienced with their immediate surroundings have a tendency to seek help from their immediate surroundings and official institutions or organizations.

### 2.2. Dependent and Independent Variables

In the study, we attempted to explain the factors affecting the behavior of women subjected to violence to seek help or not through official means. In the research's ques-

tionnaire, the women were asked whether they applied to any official institution when subjected to violence by their partners. In the second stage of the questionnaire, the women were asked to answer from which official institution (police, gendarmerie, family court, bar association, lawyer) they requested help. The questions consisted of two categories: yes and no (Yes, No).

Seven explanatory variables were used to explain formal help-seeking behavior. These variables were education level (noneducated, primary education graduate, high school, university, and above), region (West, South, Central, North, East), age groups (15–24, 25–34, 35–44, 45–59), and individual income (Yes, No). In addition, whether women subjected to violence share the abuse they experience with their immediate surroundings ("mother (Yes, No), father (Yes, No), sister (Yes, No), brother (Yes, No), friends (Yes, No), and their children (Yes, No)") was determined as an independent variable.

The region of residence reflects the heterogeneous nature of the country. The regional distribution shows differences in cultural and socioeconomic characteristics around the whole country. The western region is the most crowded, economically developed, and socially developed, whereas the eastern part of Turkey is not as developed as the western and other three parts of the country [39]. Education in Turkey (5 years plus 3 years, totaling 8 years of primary school education) is mandatory. The Istanbul Convention [40], which includes significant changes for improving women's rights, came into force in 2011. Many fundamental women's rights were guaranteed with this contract. With the Istanbul Convention, the fight against violence against women and their family members was institutionalized, and protective and preventive measures were expanded. It became possible to take immediate measures without seeking evidence and documents, and those who committed violent crimes were immediately sentenced [41]. All women and the general public in Turkey were informed of this framework of the law. In this study, the change between 2008 and 2014 was examined through the dummy variable to determine how positive the effects the Istanbul Convention and socioeconomics improvement had were on the development of the official help-seeking behavior of women subjected to violence.

## 3. Results

Table 1 shows the behavior of women subjected to violence who sought help through official means (police, gendarmerie, family court, lawyer, bar association). There are differences between the years 2008 and 2014, according to characteristics such as education, region, age, individual income, and sharing their situation with relatives. All explanatory variables were analyzed according to the significance levels of 1%, 5%, and 10% and Pearson's chi-square statistics. Generally, for all independent variables, it was found that the behavior of women subjected to violence to seek help through official means was relatively low.

**Table 1.** Formal help-seeking behavior of the proportion of women exposed to intimate partner violence (IPV). Turkey, 2008, 2014.

| Variables | | Pooled Data | | 2008 | | 2014 | |
|---|---|---|---|---|---|---|---|
| | | Seeking Help | | Seeking Help | | Seeking Help | |
| | | No | Yes | No | Yes | No | Yes |
| **Education level** | | (N = 6776) | $\chi2 = 91.7$ * | N = (4878) | $\chi2 = 52.9$ * | (N = 1898) | $\chi2 = 23.8$ * |
| **Never** attended **school** | % | 96.5 | 3.5 | 97.4 | 2.6 | 92.8 | 7.2 |
| **First- and** second-**level primary  school** | % | 92.4 | 7.6 | 94.1 | 5.9 | 88.4 | 11.6 |
| **High school** | % | 87.5 | 12.5 | 89.6 | 10.4 | 83.1 | 16.9 |
| **University and above** | % | 83.7 | 16.3 | 88.4 | 11.6 | 77.3 | 22.7 |
| **Total** | N | 6260 | 516 | 4597 | 281 | 1663 | 235 |
| | % | 92.4 | 7.6 | 94.2 | 5.8 | 87.6 | 12.4 |

**Table 1.** *Cont.*

| Variables | | Pooled Data | | 2008 | | 2014 | |
|---|---|---|---|---|---|---|---|
| | | Seeking Help | | Seeking Help | | Seeking Help | |
| | | No | Yes | No | Yes | No | Yes |
| **Region5** | | (N = 6778) | $\chi2 = 45.3$ * | (N = 4880) | $\chi2 = 16.2$ * | (N = 1898) | $\chi2 = 21.7$ * |
| **West** | % | 90.4 | 9.6 | 92.6 | 7.4 | 87.1 | 12.9 |
| **South** | % | 91.2 | 8.8 | 92.5 | 7.5 | 88.0 | 12.0 |
| **Central** | % | 91.2 | 8.8 | 93.8 | 6.2 | 84.8 | 15.2 |
| **North** | % | 91.6 | 8.4 | 94.6 | 5.4 | 83.3 | 16.7 |
| **East** | % | 95.5 | 4.5 | 95.9 | 4.1 | 93.8 | 6.2 |
| **Total** | N | 6262 | 516 | 4599 | 281 | 1663 | 235 |
| | % | 92.4 | 7.6 | 94.2 | 5.8 | 87.6 | 12.4 |
| **Age group** | | (N = 6778) | $\chi2 = 3.7$ | (N = 4880) | $\chi2 = 7.1$ | (N = 1898) | $\chi2 = 2.8$ |
| **15–24** | % | 91.6 | 8.4 | 92.7 | 7.3 | 87.0 | 13.0 |
| **25–34** | % | 93.0 | 7.0 | 95.2 | 4.8 | 86.8 | 13.2 |
| **35–44** | % | 91.8 | 8.2 | 94.0 | 6.0 | 86.6 | 13.4 |
| **45–54** | % | 92.2 | 7.8 | 93.4 | 6.6 | 89.4 | 10.6 |
| **55–59** | % | 93.5 | 6.5 | 95.2 | 4.8 | 89.3 | 10.7 |
| **Total** | N | 6262 | 516 | 4599 | 281 | 1663 | 235 |
| | % | 92.4 | 7.6 | 94.2 | 5.8 | 87.6 | 12.4 |
| **Personalincome** | | (N = 6772) | $\chi2 = 66.3$ * | (N = 4874) | $\chi2 = 30.1$ * | (N = 1898) | $\chi2 = 23.9$ * |
| **No** | % | 93.8 | 6.2 | 95.1 | 4.9 | 89.9 | 10.1 |
| **Yes** | % | 87.4 | 12.6 | 90.5 | 9.5 | 81.5 | 18.5 |
| **Total** | N | 6256 | 516 | 4593 | 281 | 1663 | 235 |
| | % | 92.4 | 7.6 | 94.2 | 5.8 | 87.6 | 12.4 |
| **Sharing with mother** | | (N = 6761) | $\chi2 = 341.9$ * | (N = 4586) | $\chi2 = 187.3$ * | N = (1898) | $\chi2 = 138.1$ * |
| **No** | % | 95.4 | 4.6 | 96.5 | 3.5 | 92.5 | 7.5 |
| **Yes** | % | 80.7 | 19.3 | 85.0 | 15.0 | 71.5 | 28.5 |
| **Total** | N | 6249 | 512 | 4586 | 277 | 1663 | 235 |
| | % | 92.4 | 7.6 | 94.3 | 5.7 | 87.6 | 12.4 |
| **Sharing with father** | | (N = 6761) | $\chi2 = 253.2$ * | (N = 4863) | $\chi2 = 159$ * | (N = 1898) | $\chi2 = 98.4$ * |
| **No** | % | 94.0 | 6.0 | 95.6 | 4.4 | 89.9 | 10.1 |
| **Yes** | % | 75.9 | 24.1 | 80.8 | 19.2 | 63.5 | 36.5 |
| **Total** | N | 6249 | 512 | 4586 | 277 | 1663 | 235 |
| | % | 92.4 | 7.6 | 94.3 | 5.7 | 87.6 | 12.4 |
| **Sharing with sister** | | (N = 6761) | $\chi2 = 69.3$ * | (N = 4863) | $\chi2 = 31.3$ * | (N = 1898) | $\chi2 = 33.3$ * |
| **No** | % | 93.8 | 6.2 | 95.2 | 4.8 | 89.9 | 10.1 |
| **Yes** | % | 87.0 | 13.0 | 90.5 | 9.5 | 79.3 | 20.7 |
| **Total** | N | 6249 | 512 | 4586 | 277 | 1663 | 235 |
| | % | 92.4 | 7.6 | 94.3 | 5.7 | 87.6 | 12.4 |

**Table 1.** *Cont.*

| Variables | | Pooled Data | | 2008 | | 2014 | |
|---|---|---|---|---|---|---|---|
| | | Seeking Help | | Seeking Help | | Seeking Help | |
| | | No | Yes | No | Yes | No | Yes |
| **Sharing with brother** | | (N = 6761) | χ2 = 117.7 * | (N = 4863) | χ2 = 56.1 * | (N = 1898) | χ2 = 67.7 * |
| **No** | % | 93.3 | 12.4 | 94.9 | 5.1 | 89.1 | 10.9 |
| **Yes** | % | 78.4 | 21.6 | 84.4 | 15.6 | 61.9 | 38.1 |
| **Total** | N | 6249 | 512 | 4586 | 277 | 1663 | 235 |
| | % | 92.4 | 7.6 | 94.3 | 5.7 | 87.6 | 12.4 |
| **Sharing withchildren** | | (N = 6761) | χ2 = 70.6 * | (N = 4863) | χ2 = 44.7 * | (N = 1898) | χ2 = 32.2 * |
| **No** | % | 93.0 | 7.0 | 94.7 | 5.3 | 88.6 | 11.4 |
| **Yes** | % | 78.5 | 21.5 | 84.8 | 15.2 | 60.3 | 39.7 |
| **Total** | N | 6249 | 512 | 4586 | 277 | 1663 | 235 |
| | % | 92.4 | 7.6 | 94.3 | 5.7 | 87.6 | 12.4 |
| **Sharing with friends** | | (N = 6761) | χ2 = 46.1 * | (N = 4863) | χ2 = 35.9 * | (N = 1898) | χ2 = 9.9 * |
| **No** | % | 93.4 | 6.6 | 95.2 | 4.8 | 88.7 | 11.3 |
| **Yes** | % | 87.4 | 12.6 | 89.7 | 10.3 | 82.5 | 17.5 |
| **Total** | N | 6249 | 512 | 4586 | 277 | 1663 | 235 |
| | % | 92.4 | 7.6 | 94.3 | 5.7 | 87.6 | 12.4 |

Source: National Survey of Domestic Violence against Women in Turkey data, 2008 and 2014. * $p < 0.01$

Formal help-seeking behavior differs according to the education level of women subjected to violence. The increase in the level of education increases formal help-seeking behavior in both 2008 and 2014. The level of formal help-seeking behavior for women subjected to violence in 2008 and 2014 is the highest at the university and higher education level. This rate was 11.6% in 2008 and 22.7% in 2014. The lowest rates for the level of formal help-seeking behavior were found for women with no education (2.6% for 2008, 7.2% for 2014). In general, as the education level increases for 2008 and 2014, formal help-seeking behavior also increases.

Turkey's western region is the most developed of the five regions, whereas the northern, southern, and central regions can be considered advanced at the same level [39]. Formal help-seeking behavior in the western region was 7.4% in 2008 and 12.9% in 2014. The lowest formal help-seeking behavior in the eastern region was 4.1% in 2008 and 6.1% in 2014. In general, formal help-seeking behavior is similar in the northern, southern, and central Anatolian regions. There was an absolute increase in formal help-seeking behavior for all regions in 2014 compared to 2008.

Regarding formal help-seeking behavior for different age groups, the general tendency for age groups to present formal help-seeking behavior decreased as the age category increased. In addition, formal help-seeking behavior in 2014 shows an increase in all categories compared to 2008. However, the relationship between formal help-seeking behavior and age groups is not significant according to Pearson's chi-square statistics.

In studies conducted in Turkey, it was concluded that women with a personal income saw less violence [42,43]. Likewise, women with a personal income are more likely to seek official help when subjected to violence than those without a personal income. In 2008, the help-seeking behavior of women without a personal income was 4.9%, whereas the behavior of women with a personal income to seek help through official means was 9.5%. In 2014, although 10.1% of women with no personal income were officially seeking help, it was found that 18.5% of women with a personal income were also officially seeking help. It can be said that women with a personal income seek more official help when exposed

to violence. In addition, there is approximately twice the increase in formal help-seeking behavior in 2014, regardless of whether the women have a personal income or not.

As mentioned in the introduction, in many societies, women subjected to violence prefer to hide the violence they experience due to sociocultural reasons. In our study, we tried to examine the effect of violence on women's help-seeking behavior by official means, especially when they share their experiences with family members or close friends. According to the results, when women shared this situation with their relatives when subjected to violence from their partners, there was a significant increase in their help-seeking behavior through official means. For example, in 2008, although the official help-seeking behavior of women who did not share their experiences with violence from their partners with their mothers and fathers was 3.5% and 4.4%, respectively, it was found that 15% and 19.2% of those who shared their experiences with their parents were seeking help officially. Although the behavior of those who did not share their experiences with their mother and father for 2014 was 7.5%, 10.1% were seeking help officially, and it was found that 28.5% and 36.5% of those who shared their experiences with their parents were officially seeking help.

As mentioned in the introduction, in many societies, women subjected to violence prefer to hide the violence they experience due to sociocultural reasons. In this study, we tried to determine the effects of women subjected to violence who share the violence they experience with family members or close friends on their formal help-seeking behavior. According to the results, when women shared the violence they experienced from their partners with their relatives, there was a significant increase in their formal help-seeking behavior. For example, in 2008, the formal help-seeking behavior of women who did not share the violence they experienced with their mother and father was 3.5% and 4.4%, respectively, whereas the formal help-seeking behavior of those who shared their experiences with their mother and father was found to be 15% and 19.2%. In 2014, the formal help-seeking behavior of women who did not share the violence they experienced with their mother and father was 7.5% and 10.1%, respectively, whereas the formal help-seeking behavior of those who shared their experiences with their mother and father was found to be 28.5% and 36.5%.

The logistic regression estimation results are shown in Table 2. The outcomes taken from the logistic regression models show that formal help-seeking behavior is positively affected by women's education, personal income, their region of residence, and whether they share their experiences with relatives. At the same time, the formal help-seeking behavior among women subjected to violence increased significantly in 2014 compared to 2008.

**Table 2.** Results of the logistic regression model applied to formal institutions to cope with IPV.

| Dependent Variable, (Formal Help-Seeking Behavior) | | | | | |
|---|---|---|---|---|---|
| Independent variable | OR | Std. Err. | Z value | Probability | [95% C.I] |
| **Education** | | | | | |
| **Never** attended **school** | 1.00 | | | | |
| **First-** and second-**level primary school** | 1.61 * | 0.2709013 | 2.85 | 0.004 | [1.16, 2.24] |
| **High school** | 2.22 * | 0.4535131 | 3.94 | 0.000 | [1.49, 3.32] |
| University **and above** | 2.29 * | 0.5703686 | 3.34 | 0.001 | [1.40, 3.73] |
| **Region** | | | | | |
| **East** | 1.00 | | | | |
| **West** | 1.29 *** | 0.1929621 | 1.71 | 0.088 | [0.96, 1.73] |
| **South** | 1.49 ** | 0.2899663 | 2.08 | 0.037 | [1.02, 2.18] |

**Table 2.** *Cont.*

| Dependent Variable, (Formal Help-Seeking Behavior) | | | | | |
|---|---|---|---|---|---|
| **Central** | 1.51 * | 0.2209161 | 2.85 | 0.004 | [1.13, 2.01] |
| **North** | 1.27 | 0.2285267 | 1.33 | 0.182 | [0.89, 1.80] |
| **Age group** | | | | | |
| **15–24** | 1.00 | | | | |
| **25–34** | 0.84 | 0.1518517 | −0.96 | 0.339 | [0.59, 1.19] |
| **35–44** | 0.97 | 0.1784865 | −0.12 | 0.906 | [0.68, 1.39] |
| **45–59** | 0.98 | 0.1896735 | −0.09 | 0.930 | [0.67, 1.43] |
| **15–24** | 0.81 | 0.1999704 | −0.83 | 0.407 | [0.50, 1.31] |
| **Year** | | | | | |
| **2008** | 1.00 | | | | |
| **2014** | 2.09 * | 0.2107253 | 7.32 | 0.000 | [1.71, 2.54] |
| **Personal income** | | | | | |
| **No** | 1.00 | | | | |
| **Yes** | 1.66 * | 0.1820482 | 4.64 | 0.000 | [1.34, 2.06] |
| **Sharing with mother** | | | | | |
| **No** | 1.00 | | | | |
| **Yes** | 3.17 * | 0.3945309 | 9.29 | 0.000 | [2.48, 4.04] |
| **Sharing with father** | | | | | |
| **No** | 1.00 | | | | |
| **Yes** | 1.72 * | 0.2522346 | 3.76 | 0.000 | [1.29, 2.30] |
| **Sharing with sisters** | | | | | |
| **No** | 1.00 | | | | |
| **Yes** | 0.96 | 0.1156339 | −0.32 | 0.749 | [0.76, 1.21] |
| **Sharing with brothers** | | | | | |
| **No** | 1.00 | | | | |
| **Yes** | 1.40 ** | 0.2271506 | 2.13 | 0.034 | [1.02, 1.93] |
| **Sharing with children** | | | | | |
| **No** | 1.00 | | | | |
| **Yes** | 2.51 * | 0.4742239 | 4.89 | 0.000 | [1.73, 3.63] |
| **Sharing with friends** | | | | | |
| **No** | 1.00 | | | | |
| **Yes** | 1.21 | 0.1437540 | 1.61 | 0.108 | [0.95, 1.52] |
| **Constant** | 0.01 | 0.0036856 | −17.92 | 0.000 | [0.01, 0.02] |
| **$R^2$** | | | | 0.24 * | |

Source: National Survey of Domestic Violence against Women in Turkey data, 2008 and 2014, significant value * $p < 0.01$ ** $p < 0.05$ *** $p < 0.10$, OR = odds ratio; C.I = confidence interval Std. Err. = standard error.

According to women with no education, help-seeking behavior by official means is 1.6 times higher in women who have received first- and second-level primary school education. Help-seeking behavior by official means is 2.22 times higher in the high-school-educated group and 2.29 times higher in those with a university- and higher-level education. The group with the highest formal help-seeking behavior is women with a university-level education and above. It was determined that the increase in education level positively increased formal help-seeking behavior.

From the perspective of regions, levels of formal help-seeking behavior are higher in each region than in the eastern part of Turkey. This situation is expected. As mentioned earlier in this article, the eastern region is the least developed region in Turkey. According to the eastern region, formal help-seeking behavior is 1.29 times higher in the western, 1.49 times in the southern, 1.51 times in the central, and 1.27 times in the northern regions. Of all these five regions, only the northern region is not statistically significant compared to the eastern region.

According to the women who do not share their experiences with domestic violence with their family, those who share their experiences with their mothers 3.17 times, those who share their experiences with their fathers 1.72 times, those who share their experiences with their sisters 0.96 times, those who share their experiences with their brothers 1.40 times, those who share their experiences with their children 2.51 times, and those who share their experiences with friends 1.21 times have shown more formal help-seeking behavior. However, the increase in formal help-seeking behavior among women subjected to violence who shared their experiences with violence with their sisters and friends is not statistically significant. According to the results obtained, it can be said that family members and close friends of women subjected to violence encourage women to formal help-seeking behavior by encouraging women exposed to violence. The formal help-seeking behavior of women who share their experiences with violence with their family members and friends is 2.09 times more in 2014 than 2008.

## 4. Discussion

According to the results of the study, the formal help-seeking behavior of women exposed to IPV who have a university education or above is 2.29 times more than women who have any form of education. Education is an essential factor in the formal help-seeking behavior of women subjected to violence. Increasing the level of education of women should be encouraged and supported to prevent violence against women and improve formal help-seeking behavior. In addition, the formal help-seeking behavior of women with a personal income was found to be 1.66 times higher than women without a personal income. The most important reasons why women exposed to IPV continue to have relationships despite being subjected to violence are due to being economically dependent on their spouse and having no shelter [44,45]. Ergöçmen et al. found that women exposed to IPV in Turkey display formal help-seeking behavior, especially when the violence they experience reaches an unbearable level [33]. For women exposed to violence in Turkey, if there are no alternative locations to stay and they have no economic power, there are only 143 shelters with a 3444-person capacity [46]. It can be said that this capacity in Turkey, where domestic violence is so widespread, is insufficient.

In this study, it was found that the development of the sociocultural region in which women live has an effect on formal help-seeking behavior. In the eastern region of Turkey, which is considered less developed, women's formal help-seeking behavior is less developed than in other regions. On the other hand, it was found that those who shared experiences of the violence they were subjected to with their family and friends could seek help via formal ways than those who did not. It can be said that the family members and friends of women subjected to violence encourage them in formal help-seeking behavior.

According to the results, it was found that women subjected to violence in 2014 were 2.09 times more likely to apply formal help-seeking behavior in 2008. In Turkey, from 2008 until 2014, many legal regulations for the improvement of women's rights were established. The most important of these are the Istanbul Convention [41], signed on 11 May 2011, and Law No. 6284 [47], enacted on 8 March 2012, on the protection of the family and the prevention of violence against women. Following these legal changes, women were informed through NGOs, symposiums, and panels. The number of women informed through these activities to apply legal methods against violence increased in 2014 compared to 2008. However, in Turkey, although the formal help-seeking behavior of women increased in recent years, it is not adequate. This study includes a comparison between 2008 and

2014 [48,49] based on the most comprehensive and reliable data representing Turkey. We can express that the Istanbul Convention may have impacted our results. However, we cannot establish a causal relationship because of the insufficient time dimension in our data. There may also be other factors that affected the formal help-seeking behavior that are not mentioned in this study due to data and research limitations. For further studies, a diff-in-diff application may be helpful to explain the effects of the Istanbul Convention, which was not our focus in this paper.

## 5. Conclusions

In this study, the formal help-seeking behavior of women subjected to violence has been examined. According to the results, it was observed that formal help-seeking behavior was low in women exposed to IPV. However, when compared to 2008 and 2014, it has been determined that there is a significant increase in formal help-seeking behavior. Although there has been an increase in women's formal help-seeking behavior over the years, it cannot be said that this increase is sufficient. This is the first research to include a comparison of data between 2008 and 2014 and examine the behavior of women subjected to violence within the framework of education, age, and regional factors.

Reducing domestic or intimate partner violence is vital in terms of gender equality, especially in developing countries. Women not under pressure would be more likely to add value to and spend more time in productive fields. This may also increase women's overall happiness, which is related to overall productivity [50]. These factors could pave the way for sustainable development and need to be solved for the long term to lift the blocks on the development road.

**Author Contributions:** A.K. conceived the presented idea. A.K. and F.K. developed the theory and performed the computations. A.K. and F.K. verified the analytical methods. A.K. wrote the manuscript with support from F.K., A.K. and F.K. provided critical feedback and helped shape the research, analysis, and manuscript. A.K. and F.K. designed the model and computational framework and analyzed the data. F.K. conceived the study and was in charge of the overall direction and planning. A.K. and F.K. discussed the results and commented on the manuscript. A.K. and F.K. contributed to the design and implementation of the research, to the analysis of the results, and to the writing of the manuscript. Both authors have read and agreed to the published version of the manuscript.

**Funding:** This research received no external funding.

**Acknowledgments:** 

**Conflicts of Interest:** The authors declare no conflict of interest.

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
