# Peer review of "Barriers to Formal Help-Seeking Behavior by Battered Turkish Women According to Sociodemographic Factors"

_sustainability, doi:10.3390/su13010165_

Round 1

Reviewer 1 Report

The paper presented is of great interest. Having data and trends on IPV in Turkey is of great relevance for comparative studies. In general, the objectives are appropriate with the methodology and analysis. The discussion and conclusions are consistent.

Three issues are proposed for the improvement of the text:

1- The text refers to the importance of the Istanbul Convention from the point of view of the subjective recognition of women's rights (lines 118, 167-171, 315). It would be interesting to further develop the contents of this Convention and especially its relationship with the development of public policies.

2- The division of the territory into four regions (N,S,E,W) is very interesting. It would be useful to explain the criteria used for the grouping of provinces in each of the regions.

3- In the conclusions (line 315), reference is made to the importance of the signature by the government of Turkey of the Istanbul Convention, as it provides an impulse for training and information on violence against women. It is very difficult to argue empirically that this has meant a variation in data from 2008 to 2014, because other factors or variables not covered in the study may be involved.

Author Response

We would like to thank you for your insightful comments and suggestions. We made possible changes that were suggested and detailed the changes in the table below. Prior to your response to your comments, we want to inform you that all the revisions and improvements are highlighted in green in our manuscript's revised version. We sincerely appreciate your insightful comments on our paper. We would like to thank you again for your valuable time and insight strengthens to our article.

Yours truly,

Correspondance author on behalf of the authors.

Reviewer 2 Report

The following corrections must be made in the text:

The purpose of the study and the objectives must be separated from the introduction.

The conclusions must be separated from the discussions.

57 "same study, %12 of women", will be corrected as: same study, 12% of women

104 "Turkey's official and most comprehensive study Ergöçmen et al. by" will be corrected as: Turkey's official and most comprehensive study by Ergöçmen et al. 

157 "age groups (15-24, 25-34, 35-44, 45-59). ), individual income" will be corrected as: age groups (15-24, 25-34, 35-44, 45-59), individual income

Table 1. "Sharing Childiren" must be corrected as: Sharing with children

298 "to violence are; being" must be corrected as: to violence are: being

There are some mistakes in the bibliography that need to be corrected:

374 references older than 10 years are accepted only for the history of the studied topic

386 must be completed

387 must be completed

406 "Anadolu Psikiyatri Dergisi" must be corrected as: Anadolu Psikiyatri Dergisi

Author Response

(The authors gave the same response as above.)
